



**Distinct Impacts of El Niño-Southern Oscillation and Indian Ocean Dipole**
**on China's Gross Primary Production**
Ran Yan[1,2], Jun Wang[1,2*], Weimin Ju[1,2*], Xiuli Xing[3], Miao Yu[4], Meirong Wang[4], Jingye Tan[1,]
[2], Xunmei Wang[1, 2], Hengmao Wang[1,2], Fei Jiang[1,2]
[1]Frontiers Science Center for Critical Earth Material Cycling, International Institute for Earth System Science,
Nanjing University, Nanjing, Jiangsu 210023, China
[2]Jiangsu Provincial Key Laboratory of Geographic Information Science and Technology, Key Laboratory for Land
Satellite Remote Sensing Applications of Ministry of Natural Resources, School of Geography and Ocean Science,
Nanjing University, Nanjing, Jiangsu 210023, China
[3]Department of Environmental Science and Engineering, Fudan University, No. 2005, Songhu Road, Yangpu
District, Shanghai 200438, China
[4]Joint Center for Data Assimilation Research and Applications/Key Laboratory of Meteorological Disaster,
Ministry of Education/Joint International Research Laboratory of Climate and Environment Change (ILCEC)/
Collaborative Innovation Center ON Forecast and Evaluation of Meteorological Disasters, Nanjing University of
Information Science and Technology, Nanjing 210044, China
Corresponding author: Jun Wang (wangjun@nju.edu.cn); Weimin Ju (juweimin@nju.edu.cn)



**Abstract**
Gross primary production (GPP) stands as a crucial component in the terrestrial carbon cycle,
greatly affected by large-scale circulation adjustments. This study explores the influence of El
Niño-Southern Oscillation (ENSO) and Indian Ocean Dipole (IOD) on China's GPP, utilizing
long-term GPP data generated by the Boreal Ecosystem Productivity Simulator (BEPS). Partial
correlation coefficients between GPP and ENSO reveal substantial negative associations in
most parts of western and northern China during the September-October-November (SON)
period of ENSO development. These correlations shift to strongly positive over southern China
in December-January-February (DJF), then weaken in March-April-May (MAM) in the
following year, eventually turning generally negative over southwestern and northeastern China
in June-July-August (JJA). In contrast, the relationship between GPP and IOD basically exhibits
opposite seasonal patterns. Composite analysis further confirms these seasonal GPP anomalous
patterns. Mechanistically, we ascertain that, in general, these variations are predominantly
controlled by soil moisture in SON and JJA, but temperature in DJF and MAM. Quantitatively,
China's annual GPP demonstrates modest positive anomalies in La Niña and nIOD years, in
contrast to minor negative anomalies in El Niño and pIOD years. This results from
counterbalancing effects with significantly greater GPP anomalous magnitudes in DJF and JJA.
Additionally, the relative changes in total GPP anomalies at the provincial scale display an east-
west pattern in annual variation, while the influence of IOD events on GPP presents an opposing
north-south pattern. We believe that this study can significantly contribute to our
comprehension of how intricate atmospheric dynamics influence China's GPP on an
interannual scale.
**Key words:** Gross primary production, China, El Niño-Southern Oscillation, Indian Ocean
Dipole, BEPS

**Highlight**
(1)  Impacts of ENSO and IOD on China's GPP vary with seasons, showing nearly opposite



patterns.
(2)   Soil moisture controls GPP in fall and summer, while temperature plays a key role in

48        winter and spring.

(3)   Counterbalancing causes modest positive GPP anomalies in La Niña and nIOD,

50        contrasting with minor negative anomalies in El Niño and pIOD.


**1.Introduction**
Vegetation photosynthesis, a pivotal physiological process affecting the terrestrial carbon cycle,
predominantly governs variations in the net biome productivity (NBP), surpassing the impact
of total ecosystem respiration (Piao et al., 2020; Wang et al., 2022; Wang et al., 2018). Gross
primary production (GPP) represents the total amount of carbon dioxide assimilated by plants
per unit time through the photosynthetic processes, acting as a crucial carbon flux in mitigating
anthropogenic $CO_2$ emissions (Gough, 2012; Houghton, 2007). However, despite evident long-
term increasing trends in GPP, primarily attributed to $CO_2$ fertilization (Ryu et al., 2019;
Schimel et al., 2015; Yang et al., 2022), it also shows regional and global interannual variations.
These variations are largely linked to climate fluctuations driven by ocean-atmosphere
interactions and the teleconnections (Wang et al., 2021b; Ying et al., 2022). To date, the impact
of such teleconnections on China's GPP remains insufficiently documented.

The El Niño-Southern Oscillation (ENSO) exerts a significant influence on the global terrestrial
carbon cycle, which is the dominant mode of inter-annual climate variability (Bauch, 2020;
Kim et al., 2017; Wang et al., 2016; Wang et al., 2018; Zeng et al., 2005). Within this context,
GPP typically assumes a leading role in shaping the response of terrestrial carbon sinks to
ENSO events (Ahlstrom et al., 2015; Wang et al., 2018; Zhang et al., 2018). Global patterns
reveal a negative GPP anomaly of approximately −1.08 Pg C yr$^{-1}$ during El Niño years,
contrasting a positive GPP anomaly of about 1.63 Pg C yr$^{-1}$ in La Niña years (Zhang et al.,
2019). However, the impact of ENSO on GPP exhibits significant regional differences. At





present, while existing researches have predominantly focused on the response of tropical GPP
to ENSO, studies specific to China are relatively limited. Liu et al. (2014) highlighted the effects
of ENSO on crop growth in the North China, and Li et al. (2021) demonstrated that the response
of GPP to El Niño varies with the phase of the Pacific Decadal Oscillation (PDO) in the eastern
China.

ENSO is not the sole global climatic oscillation, influencing the terrestrial carbon cycle.
Another significant player is the Indian Ocean Dipole (IOD), a tropical coupled ocean-
atmosphere mode (Saji et al., 1999), which also affects the terrestrial carbon cycling by
modulating the climate circulations (Wang et al., 2022; Wang et al., 2020; Wang et al., 2021b;
Yan et al., 2023). Research indicates that IOD events can influence precipitation in China, with
effects lasting from the year of the event through the subsequent summer (Zhang et al., 2022a).
Zhang et al. (2022b) also proved that extreme pIOD events in 2019 affected the precipitation in
summer 2020 in Eastern China, and proposed that the summer precipitation in the following
year was mainly affected by IOD in northern China, while by ENSO in the Yangtze River Basin.
Additionally, a prior study explored the influence of the extreme positive IOD (pIOD) event in
2019 on GPP anomalies across the Indian Ocean rim countries. It suggested a conspicuous
negative GPP anomaly occurred in eastern China during the September-October-November
(SON) (Wang et al., 2021b).

The primary objective of this study was to comprehensively assess the impact of ENSO and
IOD events on GPP in China. To this end, we initially employed partial correlation analysis to
elucidate the relationship between GPP and climate anomalies, specifically soil moisture and
temperature, induced by ENSO and IOD events across various seasons. The analysis utilized
historical long-term GPP data spanning from 1981 to 2021, simulated by the Boreal Ecosystem
Productivity Simulator (BEPS) model. The aim was to get a preliminary understanding of the
influence exerted by ENSO and IOD. Furthermore, composite analysis was adopted to illustrate
the actual responses during distinct events, including individual ENSO and IOD occurrences.



The ensuing discussion will delve into the analysis results on national, regional, and provincial
scales.

**2.Datasets and methods**
**2.1 Datasets used**
The sea surface temperature (SST) dataset are derived from the Monthly NOAA's Extended
Reconstructed Sea Surface Temperature version 5 (ERSSTv5) (Muñoz, 2019). It is generated
on a 2°x2° grid, using statistical methods to enhance spatial completeness. Commencing from
January 1854 to the present, the monthly SST data includes anomalies computed with respect
to a 1971-2000 monthly climatology.

Meteorological data were adopted from ERA5-Land monthly averaged data with $0.1° \times 0.1°$
grids, including 2m surface air temperature (TAS), and volumetric soil moisture (SM) during
the period from 1981 to 2021. ERA5-Land was created by replaying the land component of the
ECMWF ERA5 climate reanalysis at a higher resolution compared to ERA5. Reanalysis
combines model data with global observations into a consistent dataset based on the laws of
physics. The original soil moisture data was divided into four layers based on different surface
depths. These layers were depth-weighted and then aggregated into the average soil moisture
to a depth of 289cm ($m^3 m^{-3}$).

GPP spanning from 1981 to 2021 was simulated by the BEPS model, featuring a horizontal
resolution of $0.0727° \times 0.0727°$. The BEPS model, originally developed for Canadian boreal
ecosystems, has been re-constructed for GPP simulations on the global scale (Chen et al., 1999;
Chen et al., 2012). BEPS is a process-based model driven by satellite-observed leaf area index
(LAI), meteorological data, land cover types, soil texture, and $CO_2$ concentration to simulate
the daily carbon flux of terrestrial ecosystems (Chen et al., 2019; Liu et al., 1997). The input



data used to drive GPP in this study include ERA5 meteorological data (Hersbach et al., 2023),
GLOBMAP LAI product (Liu et al., 2012), Land Cover Classification System (LCCS)
generated by the Food and Agriculture Organization (FAO) of the United Nations (Friedl and
Sulla-Menashe, 2019), Harmonized World Soil Database v1.2 from FAO (Fischer et al., 2008),
and $CO_2$ concentration based on the Global Monitoring Laboratory from NASA (Lan et al.).
Notably, BEPS distinguishes itself from other models through the organic combination of
remote sensing data and mechanistic modelling. It produces simulation datasets for GPP, Net
primary productivity (NPP) and evapotranspiration (ET). Key features of BEPS include the
incorporation of sunlit-shaded leaf stratification strategy (Norman, 1982). The model calculates
canopy-level photosynthesis by summing the GPP of sunlit and shaded leaves (Chen et al.,

137    1999).

$$GPP = A_{sun}LAI_{sun} + A_{shade}LAI_{shade} \qquad (1)$$
$$LAI_{sun} = 2\cos\theta \left[1 - exp\left(-\frac{0.5\Omega LAI}{\cos\theta}\right)\right] \qquad (2)$$
$$LAI_{shade} = 1 - LAI_{sun} \qquad (3)$$
where $A_{sun}$ and $A_{shade}$ represent the amount of photosynthesis at per sunlit and shaded leaf,
respectively; $LAI_{sun}$ and $LAI_{shade}$ represent the canopy-level sunlit and shaded LAI,
respectively; $\Omega$ is the foliage clumping index indicaiting the influence of foliage clustering on
radiation transmission, and $\theta$ is the solar zenith angle.
The accuracy of carbon flux products simulated by BEPS has been validated in previous studies
(Chen et al., 2019; He et al., 2021). We also used the measured site data from ChinaFlux
(http://chinaflux.org/) and National Tibetan Plateau Third Pole Environment (Li et al., 2013)
(Table S1) to assess the performance of BEPS simulated GPP (Fig. S1). Our analysis reveals a
high consistency between simulated and observed GPP, with an average $R^2$ of 0.77 ($p < 0.05$)
and an average root mean square error (RMSE) of 1.70 gC m$^{-2}$ day$^{-1}$. In addition, the global
terrestrial GPP from FluxSat product Version 2.2 (Joiner et al., 2018) was also used to assess
the reliability of BEPS GPP. FluxSat GPP is obtained by using light-use efficiency (LUE)
framework based on Moderate-resolution Imaging Spectroradiometer (MODIS) satellite data,
eliminating the dependency on other meteorological input data. The comparison between BEPS





GPP and FluxSat GPP data revealed a robust agreement, with a correlation coefficient ($r$) of
0.63 ($p < 0.05$) and a RMSE of 1.1 Pg C yr$^{-1}$ (Fig. S2). These consistencies underscore the
reliability of the BEPS GPP data in capturing terrestrial carbon flux dynamics.

**2.2 Anomaly calculation**

To calculate anomalies, we initially eliminated the long-term climatology to get rid of the
seasonal cycle. Subsequently, we subtracted the 7-year running average for each grid to
eliminate the decadal oscillation and long-term trends for all the variables. Further, refinement
involved smoothing the derived GPP and climate anomalies using a 3-month running average
to remove the intra-seasonal variability. For consistency, the BEPS simulated GPP data was
resampled to 0.1° × 0.1°. To align with this, non-vegetated areas in the climate data were
masked according to the resampled BEPS GPP, uniformity in spatial representation.

**2.3 Definition of climate events**

The Oceanic Niño Index (ONI) is used to define ENSO events (Fig. 1a), which represents the
3-month running mean SST anomaly in the Niño 3.4 region (5°N-5°S, 120°-170°W). The
positive phase of an ENSO event (El Niño) is characterized by the ONI exceeding +0.5K for
five consecutive overlapping 3-month periods. Conversely, the negative phase of an ENSO
event (La Niña) occurs when the ONI is below −0.5K for five consecutive overlapping 3-month
periods. The severity of the event can be further categorized into weak (0.5~0.99), moderate
(1.00~1.49), strong (1.50~1.99) and extremely strong (≥2.00) based on the absolute value of
the ONI. To qualify for a specific rating, an event should meet or exceed a threshold for at least
three consecutive overlapping three-month periods.

Moreover, the Dipole Mode Index (DMI) is employed to identify IOD events (Saji et al., 1999).
The DMI is calculated from SST differences between the Western Equatorial Indian Ocean
(10°S-10°N, 50°-70°E) and the South-eastern Equatorial Indian Ocean (10°S-0°N, 90°-110°E)
(Fig.1b). Given that the short duration of IOD events with a tendency to peak during the SON,



the standard deviation of SON DMI (0.52K from 1981 to 2021) is used as the criterion for
identifying IOD events. A positive phase IOD (pIOD) event is defined when the absolute value
of DMI is greater than or equal to one standard deviation (0.52 K) for three consecutive 3-
month periods. Additionally, a strong pIOD event is identified if the DMI value exceeds two
standard deviations (1.04 K).

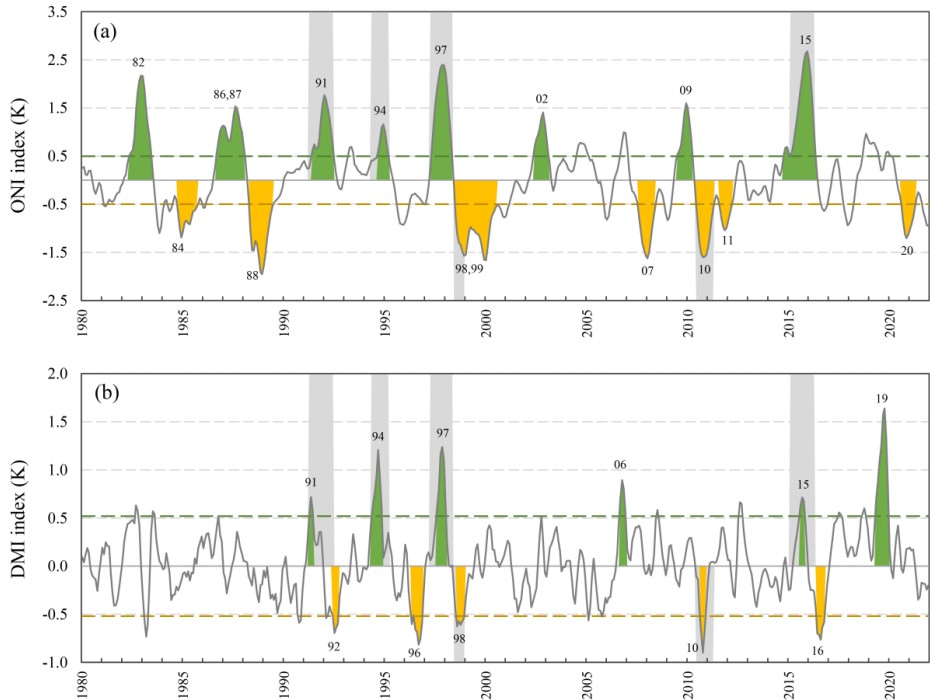


Fig.1 Time series of the Oceanic Niño Index (ONI) (a) and the Dipole Mode Index (DMI) (b) from 1980
to 2022. The positive phase events (El Niño and positive Indian Ocean Dipole (pIOD)) are filled in
green and the negative phase events (La Niña and negative IOD (nIOD)) are filled in yellow, and the
events are also labeled with a two-digit year. The green and yellow dashed lines represent the positive
and negative thresholds for El Niño-Southern Oscillation (ENSO) and IOD, respectively. The gray
background indicates years with the simultaneous ENSO and IOD events.
**2.4 Partial correlation analysis**
To comprehensively assess the impacts of ENSO and IOD on GPP, while accounting for the



influence of other events, partial correlation analysis (pcor) was employed, following the
previous studies (Saji and Yamagata, 2003; Wang et al., 2021b). The definition of *pcor* for *x*
and *y*, controlling for *z*, is given by:
$$pcor_{yx.z} = \frac{r_{yx} - r_{yz}r_{xz}}{\sqrt{1 - r_{yz}^2}\sqrt{1 - r_{xz}^2}} \tag{4}$$

where $r_{yx}$ is the correlation of the dependent variable *y* and the explanatory variable *x* (e.g.,
DMI), and the same is for $r_{yz}$ and $r_{yx}$. The two-tailed Student's *t*-test was used to calculate
the statistical significance of each pixel result:
$$t = pcor_{yx.z}\sqrt{\frac{n - 2 - k}{1 - pcor_{yx.z}^2}} \tag{5}$$

where *n* and *k* are the number of samples and conditioned variables, respectively.

**2.5 Composite analysis**
When enumerating the years of ENSO and IOD events, we retained all the years of IOD events
and ENSO events of above the moderate intensity. Individual events and compound events were
categorized and summarized in Table 1. In this study, a compound event refers to the
simultaneous occurrence of ENSO and IOD, primarily El Niño & pIOD and La Niña & negative
IOD (nIOD). IOD typically peaked in the September-October-November (SON, yr0), while
ENSO peaked in the December(yr0)-January(yr1)-February(yr1) (DJF), and the influence of
the two events could extend until the summer of the following year. Therefore, we selected four
seasons from SON to June-July-August (JJA) in the following year for composite analysis in
this study. In addition, the year 1991 was excluded due to the strong eruption of Mount Pinatubo,
which had a large impact on the global carbon cycle (Mercado et al., 2009).

**Table 1**. Occurrences of ENSO and IOD events from 1981 to 2021.

| Events | Years |
|---|---|
| El Niño | 1982, 1986, 1987, 2002, 2009 |
| La Niña | 1984, 1988, 1999, 2007, 2011, 2020 |



| pIOD | 2019 |
| nIOD | 1992, 1996, 2016 |
| El Niño & pIOD | 1994,1997, 2015 |
| El Niño & nIOD | - |
| La Niña & pIOD | - |
| La Niña & nIOD | 1998, 2010 |




**3.Results**
**3.1 Historical relationship between GPP and ENSO**



Fig. 2 Spatial patterns of partial correlation coefficients (*pcor*) between ONI and gross primary productivity (GPP) (a-d), surface air temperature (TAS) (e-h), soil moisture (SM) (i-l) in different seasons, controlling for the effect of DMI. Hatched areas represent significance at $p \leq 0.05$ based on the two-tailed Student's *t*-test. (m-p) Heatmaps represent the relationships of the *pcor* patterns among GPP, TAS, and SM, and bar charts illustrate the pattern correlations of these *pcor* values between GPP and TAS and SM on the national scale for each season. Notably, asterisks (*) in the bar charts denote significance at $p > 0.05$.







Fig. 3 Same as Fig.2, but for DMI, controlling the effect of ONI.



We analyzed the *pcor* patterns between GPP, climate anomalies, and events using long time
series data (Figs. 2 and 3). Following this, we calculated pattern correlation coefficients
between the GPP and climate *pcor* patterns (including all the pixels over China), aiming to
investigate the varying impacts of TAS and SM on photosynthesis across different seasons
(Figs. 2m-p, and 3m-p).

Figure 2 reveals notable seasonal variations in the *pcor* patterns between GPP, related climate
anomalies, and ONI index in December-January-February (DJF) when ENSO peaked,
controlling the effect of DMI in September-October-November (SON) when IOD peaked.
During SON, significant negative *pcor* between GPP and ONI is observed in regions including
the Tibetan Plateau, Southwestern China, Loess Plateau, and Liaoning province (Fig. 2a).
Clearly, this pattern aligns closely with the *pcor* pattern between soil moisture and ONI (Figs.
2a and i). The pattern correlation analysis between GPP and both TAS and SM underscores the
dominance of SM in influencing GPP anomalies, indicated by a correlation coefficient of 0.30
($p < 0.05$). This finding suggests that the soil moisture deficit induced by El Niño largely
inhibits vegetation photosynthesis during this season (Fig. 2m).

Along with the peak of ENSO events in DJF, the *pcor* pattern between GPP and ONI exhibits
a distinct shift from the pattern in SON. Notably, DJF showcases significant positive *pcor*
values over large areas in southern China and weak positive *pcor* in the North and Northeastern
China (Fig. 2b). During this period, temperature emerges as a more influential factor in driving
GPP changes, reflected in a nation-wide pattern correlation coefficient of 0.36 ($p < 0.05$) (Fig.
2n). Specifically, higher winter temperatures during El Niño, coupled with sufficient soil
moisture, contribute to a substantial enhancement in GPP across Southern China. In contrast,
the impact is weaker in the North and Northeast China due to the vegetation being in the non-
growing season, and localized soil water deficits (Figs. 2b, f, and j). In addition, GPP
experiences inhibition in some areas of southwestern China due to low temperatures and soil
drought.






Subsequently, the positive *pcor* of GPP decreases, or even turns into weak negative values from
DJF to March-April-May (MAM) in southern China. These changes are primarily attributed to
shifts of temperature, with a pattern correlation coefficient of −0.09 ($p < 0.05$) (Figs. 2c, g, and
o). Conversely, the positive *pcor* of GPP continues to increase in northern Sichuan, aligning
with the positive *pcor* of temperature (Figs. 2c and g), and in northern Hebei and parts of
neighboring Inner Mongolia, corresponding to the weak positive *pcor* of soil moisture (Figs.
2c and k).

Moving into JJA, the *pcor* of GPP exhibits widespread negative values again (Fig. 2d). In
general, during El Niño, increased soil moisture and lower temperatures greatly contribute to
enhanced GPP, while drier soil moisture and higher temperatures inhibit the increase in GPP
(Fig. 2p). Regionally, higher temperatures and lower soil moisture both contribute to the
negative GPP anomalies over southwestern China. However, lower soil moisture
predominantly curtails GPP over the Tibetan Plateau, the Yellow River basin, and northeastern
Inner Mongolia. Overall, the correlation coefficients between GPP and TAS and SM in summer
are comparable, with soil moisture exhibiting a slightly higher effect, represented by a
correlation coefficient of 0.48 ($p < 0.05$), compared to a correlation coefficient of −0.37 ($p <$
0.05) for temperature.

**3.2 Historical relationship between GPP and IOD**
In comparison, the *pcor* patterns between GPP and DMI in SON, controlling for the effect of
ONI in DJF, exhibit nearly opposite patterns to those between GPP and ONI (Figs. 2 and 3). In
detail, GPP demonstrates significant positive *pcor* values with DMI in southwestern China and
eastern Inner Mongolia, but displays significant negative *pcor* with DMI in southeastern China
during SON (Fig. 3a). In terms of climate drivers, during the pIOD events, for instance, wetter
soil and lower temperatures both benefit the significant enhancement in GPP in southwestern



China, while higher temperatures largely contribute to the enhancement in GPP over eastern
Inner Mongolia. Conversely, GPP is largely inhibited by the dry conditions in southeastern
China (Figs. 3e and i). Overall, soil moisture dominates the GPP anomaly in China, with a
correlation coefficient of 0.26 ($p < 0.05$) (Fig. 3m).

In DJF, GPP exhibits widespread significant negative *pcor* with DMI (Fig. 3b), primarily due
to the widespread negative *pcor* of temperature, characterized by a correlation coefficient of
0.04 ($p < 0.05$) (Figs. 3f and n). Moving into MAM, the significant negative *pcor* between GPP
and DMI carried on from those in DJF, but shifts to weak positive *pcor* in southeastern China,
driven by the significant positive *pcor* of temperature (Figs. 3c and g). However, the significant
negative *pcor* of soil moisture in the Jianghuai Basin and North China still negates the positive
effect of temperature (Fig. 3k). During this period, temperature remains the dominant factor,
with a nation-wide pattern correlation coefficient of 0.15 ($p < 0.05$) with GPP (Fig. 3o).

In JJA, the situation undergoes a change, showing the significant positive *pcor* of GPP over
southwestern, north and northeast China, and weak negative *pcor* over southeastern China (Fig.
3d). In other words, lower temperatures and gradually wetter soil are conducive to the increase
in vegetation photosynthesis, but heat and dry conditions cause the weak inhibition of
photosynthesis in southeastern China during the pIOD (Figs. 3p). However, unlike the ENSO
event, the role of temperature is slightly higher than that of SM in the IOD event, and the
correlations between GPP and TAS and SM are −0.39 and 0.36 ($p < 0.05$), respectively.





## 3.3 GPP anomalies caused by specific ENSO and IOD events


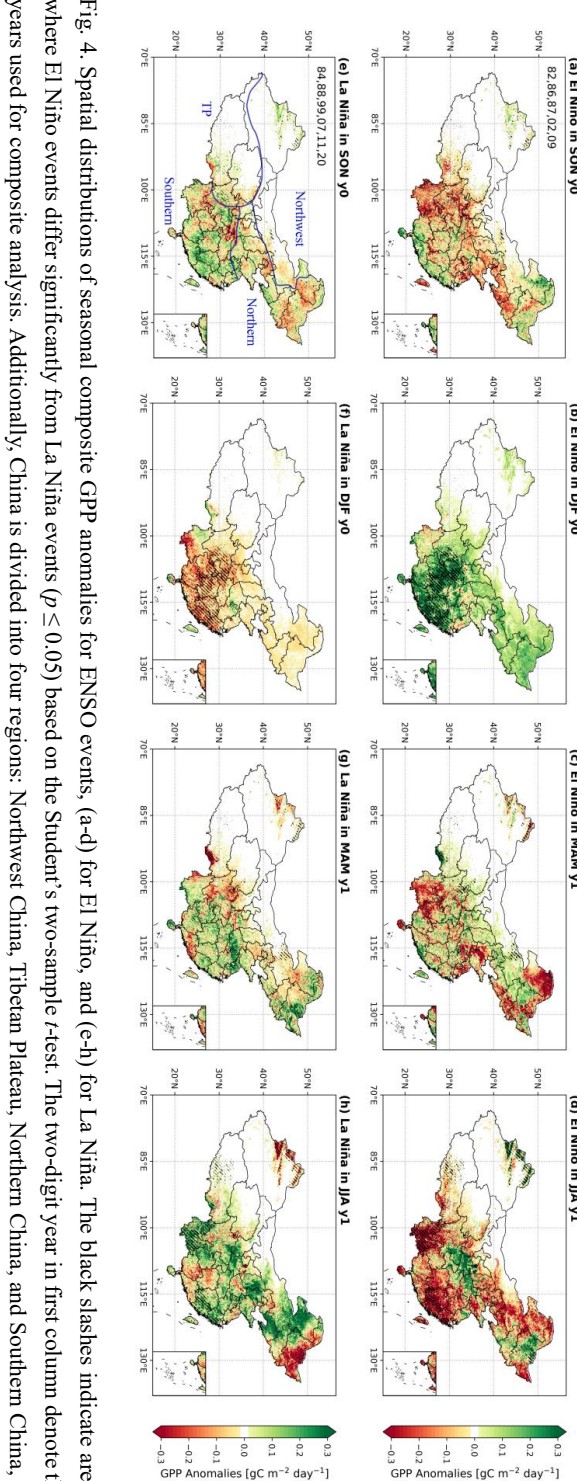


Fig. 4. Spatial distributions of seasonal composite GPP anomalies for ENSO events, (a–d) for El Niño, and (e–h) for La Niña. The black slashes indicate areas
where El Niño events differ significantly from La Niña events ($p \leq 0.05$) based on the Student's two-sample $t$-test. The two-digit year in first column denote the
years used for composite analysis. Additionally, China is divided into four regions: Northwest China, Tibetan Plateau, Northern China, and Southern China, as
shown in (e), which is used in the following context.





Fig. 5. Similar to Fig. 4, but for spatial distributions of seasonal composite GPP anomalies for IOD events, (a-d) for pIOD, and (e-h) for nIOD. We did not
conduct the significance test here owing to the limited samples.

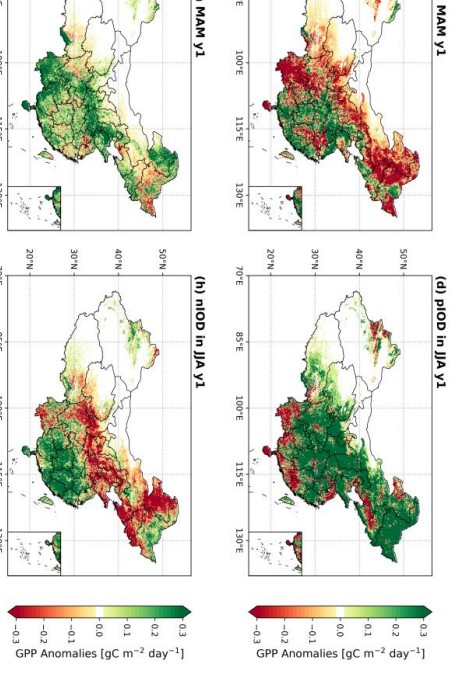



While we have elucidated the historical relationship between GPP and ENSO and IOD events
through partial correlation coefficients and discussed the underlying climate drivers, we here
specifically selected actual events to conduct a composite analysis. This approach aims to
further comprehensive understanding of the effects of ENSO and IOD events on GPP variations
in China.

### 3.3.1 ENSO-induced GPP anomalous patterns

The impacts of El Niño and La Niña events exhibit opposite influences on GPP with obvious
seasonal variations (Fig. 4). Specifically, during SON, GPP anomalies are relatively weak,
indicating some suppressions over southwestern China and north China during El Niño events,
primarily attributed to dry conditions there (Figs. 4a and S4a). As ENSO peaks in DJF, GPP is
significantly strengthened during El Niño events and suppressed during La Niña events,
especially over southern China (Figs. 4b and f), aligning well with the patterns of *pcor* between
GPP and ONI, controlling the effect of DMI (Fig. 2b). Concurrently, the widespread higher
temperatures and wetter soil moisture both contribute to enhanced GPP over southern China
during El Niño events (Figs. S3b and S4b), while colder temperatures and drier soil moisture
lead to GPP suppression there during La Niña (Figs. 2f and 3f). It is worth mentioning that GPP
shows insignificant changes over north China in DJF although soil water deficits are still severe
(Fig. S4b). This is mainly because of the non-growing season for vegetation. In MAM as ENSO
weakens and vegetation starts to grow in the extratropics, the enhanced GPP over southern
China in DJF during El Niño events diminishes, even transitioning into a notable GPP reduction
over southwestern China, north China, and northeastern China (Fig. 4c). This transition is
conspired by phenological and climate changes including colder temperatures and prolonged
dry conditions (Figs. S3c and S4c). The GPP pattern exhibits the opposite transition in La Niña
(Fig. 4g). Moving to JJA, dry and hot conditions (Fig. S3d and S4d) lead to significant negative
GPP anomalies in southeastern and southwestern China in El Niño (Fig. 4d), whereas cool and
wet conditions result in positive GPP anomalies in La Niña events (Fig. 4h). Overall, GPP



anomalies induced by ENSO events in DJF and JJA are more pronounced than those in SON
and MAM, corresponding to the life cycle of event and vegetation growth periods, respectively.
Crucially, they demonstrate distinct GPP patterns, with significant enhancements in DJF and
reductions in JJA during El Niño events and reverse during La Niña events, aligning well with
the *pcor* pattern between GPP and ONI, controlling for the effect of DMI (Fig. 4). In addition,
the effect of ENSO on vegetation in southern China appears more substantial.

**3.3.2 IOD-induced GPP anomalous patterns**
During the period from 1981 to 2021, we only find one independent but extreme pIOD event
occurred in 2019 according to our criterion (Table 1). This extreme pIOD event extended from
June to December, a longer duration compared to other IOD events. Different from ENSO,
IOD basically peaks in SON. GPP anomalies induced by this extreme event align closely with
the long-term *pcor* patterns between GPP and DMI, controlling for the effect of ONI (Fig. 3).
Specifically, significant reductions in GPP occur in southeastern China in SON (Fig. 5a),
predominantly due to heat stress and severe drought conditions (Figs. S5a and S6a), consistent
with the findings revealed by Wang et al. (2021b). In DJF, the seasonal legacy of vegetation
state (Yan et al., 2023) and prolonged droughts lead to the widespread GPP reductions (Figs.
5b and S6b), outweighing the potential positive effect of higher temperatures (Fig. S5b). Of
course, the decline of GPP in southwestern China appears linked to lower temperatures (Figs.
5b and S5b). During MAM, the mitigation of soil moisture deficit and favorable higher
temperatures in southern China facilitate a shift in GPP from decline to increase (Fig. 5c). In
the north, persistent drought conditions notwithstanding (Fig. S6c), higher temperatures and
the onset of the growing season contribute to the enhanced GPP (Fig. 5c). In JJA, increased
precipitation over the Yangtze and Yellow River basins (*Zhang et al.*, 2022) alleviates the soil
moisture deficits (Fig. S6d). Coupled with the relatively lower temperatures, this leads to
widespread GPP increases. Conversely, GPP suppressions in provinces south of 25°N and
around the Bohai Sea are attributed to higher temperatures and soil water deficits (Figs. 5d,



S5d, and S6d).

In contrast to the intense 2019 pIOD event, our composite analysis incorporates three weak
nIOD events, resulting in comparatively milder anomalies. In SON, different from pIOD event,
negative GPP anomalies in nIOD mainly appear in the provinces of Guizhou, Hunan, and
Guangxi (Fig. 5e), associated well with concurrent dry conditions (Fig. S6e). In DJF, although
the spatial pattern of soil moisture remains largely consistent with SON (Fig. S6f), a shift from
negative to positive temperature anomalies mitigates the evident GPP reductions (Fig. 5f). The
ongoing soil wetting and the onset of the growing season in northern hemisphere in MAM
result in the increased GPP over the Yellow River Basin and southwestern China (Figs. 5g, S5g,
and S6g). Subsequently, in JJA, the combination of wetter soil and lower temperatures
facilitates vegetation photosynthesis in southern China, while drier soil largely contributes to
the reduction in GPP in the north and northeastern China (Figs. 5h, S5h, and S6h).





### 3.3.3 National and regional total GPP anomalies

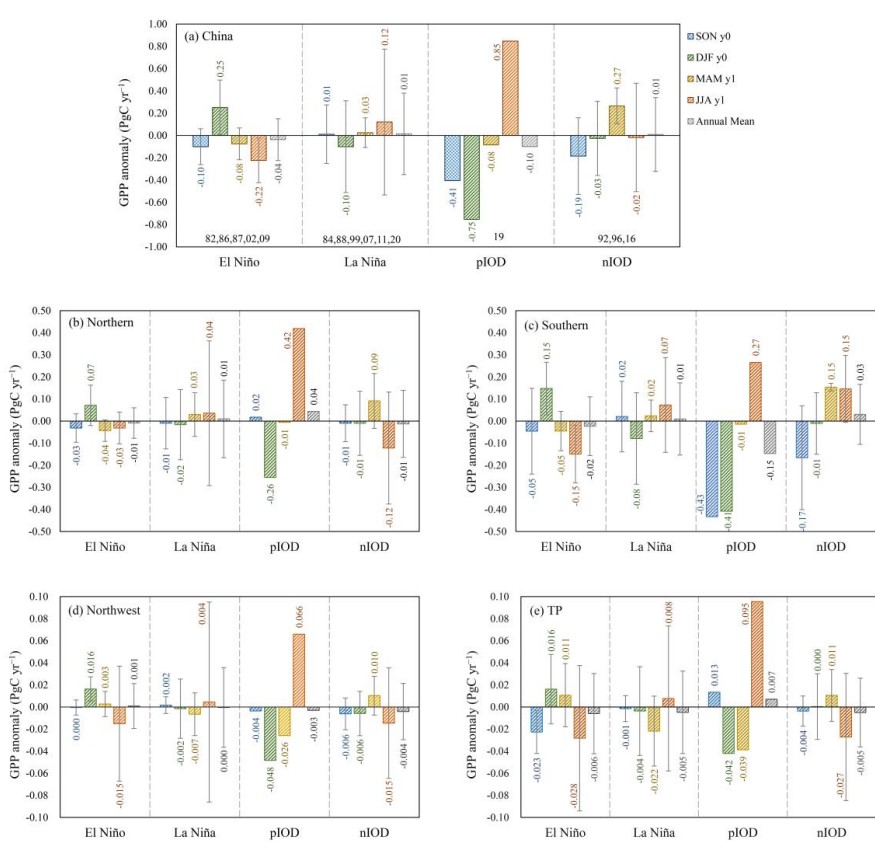

Fig. 6. The seasonal and annual mean anomaly of GPP in different classified events for China (a), for
Northern China (b), for Southern China (c), for Northwest China (d), and for Tibetan Plateau (e). The
error bars show the standard deviation of different events in the composite analysis.

We calculated the total GPP anomaly in China and various geographic regions for each
classified event on both seasonal and annual scales (Fig. 6). Regionally, the geographical
divisions include Northern China, Southern China, Northwest China, and Tibetan Plateau (Fig.
4e). Notably, the North-South boundary aligns closely with the 0° isotherm in January and the
annual precipitation line of 800 mm. The division between the North and the Northwest is



determined by the annual precipitation line of 400 mm, and the Tibetan Plateau is segmented
based on topographic factors.

In general, the GPP anomalies exhibit noticeable differences on the seasonal scale, while the
total annual anomalies do not show a significant magnitude due to the mutual offset of positive
and negative anomalies in different seasons. However, it is worth noting that our annual totals
are calculated from the SON in the developing year of the event to the JJA in the following
year. This method deviates from the traditional calendar year, and as per the conventional
definition of a "year", the annual anomalies induced by these events can indeed be substantial.

Specifically, taking a national perspective (Fig. 6a), GPP anomalies during the El Niño and La
Niña events exhibit opposite signs in DJF and JJA, with greater magnitudes during these peak
periods of the events and the most vigorous growth period of vegetation, respectively. In terms
of the development process of the event, the annual anomaly of GPP is negative during El Niño,
with a magnitude of $-0.04 \pm 0.19$ Pg C yr$^{-1}$, but positive during La Niña events, with a
magnitude of $0.01 \pm 0.37$ Pg C yr$^{-1}$. The asymmetry of the positive and negative phases of IOD
is also evident in the total anomaly. For the pIOD event in 2019, GPP shows strong negative
anomalies with values of $-0.41$ Pg C yr$^{-1}$ in SON and $-0.75$ Pg C yr$^{-1}$ in DJF. Conversely, it
exhibits a marked positive anomaly in the following JJA, with a value of 0.85 Pg C yr$^{-1}$. The
annual total of GPP anomaly is opposite for pIOD and nIOD events, showing $-0.10$ Pg C yr$^{-1}$
and $0.01 \pm 0.33$ Pg C yr$^{-1}$, respectively. Moreover, large standard deviation indicated that there
are large uncertainties in the impact of different events, and each event has its uniqueness
(Capotondi et al., 2015).

Additionally, the variation of GPP anomaly in each region is basically consistent with that at
the national scale, especially in the Southern. But regional differences indeed exist in the total
amount of GPP anomalies, demonstrating the difference in the impact of events on different
regions' GPP. Taking the 2019 extreme pIOD event as an example, the GPP showed a



significant negative anomaly in the Southern during the SON (Fig. 6c), resulting in negative
anomalies in GPP at the national scale (Fig. 6a), but weak positive anomalies in the Northern
and TP (Figs. 6b and e). Then, the GPP anomaly was close to zero in the Northern and Southern
in MAM (Figs. 6b and c), while it was still a significant negative anomaly in the Northwest
and TP (Figs. 6d and e). Moreover, the negative annual GPP anomalies in the Southern and
Northwest offset the positive anomalies of the TP and Northern, making a negative annual GPP
anomaly in the national of this event.

In terms of the magnitude of GPP anomalies, they are more pronounced in the Northern and
Southern regions, characterized by lusher vegetation, mostly less than 0.5 Pg C yr$^{-1}$.
Meanwhile, GPP anomalies are relatively weaker in the Northwest and TP regions, primarily
covered by grassland, generally less than 0.1 Pg C yr$^{-1}$. Further, we calculate the contributions
of different regions to the national total GPP anomaly in each event (Table S3), referencing an
index described in the article by Ahlstrom et al. (2015), as detailed in the supplementary method.
Overall, the GPP anomaly in the Southern region dominates the national GPP variation,
contributing approximately 68% to ENSO events and 46% to IOD events, respectively. The
Northern GPP anomaly contributes approximately 28% to the national GPP variation in ENSO
events and 39% in IOD events. In addition, the contribution of GPP anomaly in the Northwest
and TP regions to the national GPP variation is within 10%.



### 3.3.4 Relative changes in total GPP anomalies at provincial scale



Fig. 7. Spatial distributions of relative changes of total composite anomalies of GPP at provincial scale for different classified events.

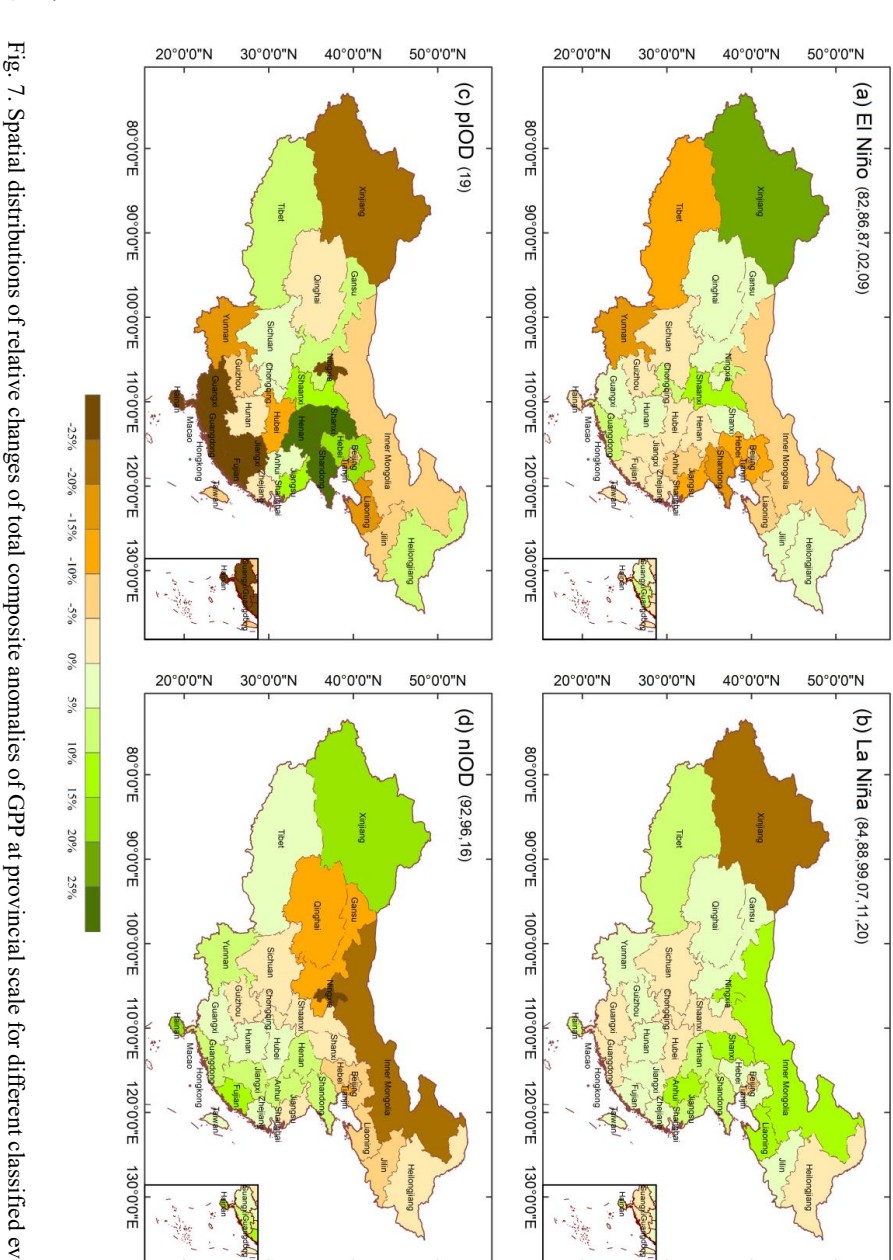



We presented the spatial patterns of mean GPP anomalies from the SON in the developing year
to the JJA in the decaying year (Fig. S7) and further calculated provincial total GPP anomalies
(Fig. S8 and Table S3). Provinces with more extensive forest coverage, such as Yunnan, central
provinces housing the Qinling Mountains, and northeast provinces where the Greater and
Lesser Hinggan Mountains are situate, exhibit relatively larger provincial GPP anomalies.
However, differences are apparent among different events (Fig. S8). Considering differences
in area and vegetation coverage across provinces, our focus centers on the relative change of
GPP anomalies (Fig. 7). It's important to note that, due to different years used in composite
analysis, our quantitative comparisons are limited to the same event within different provinces,
while qualitative descriptions are extended to different events.
El Niño events generally induce substantial GPP changes in two main regions with a relative
change of over 10% (Fig. 7a). One region encompasses the northern coastal provinces,
including Tianjin, Hebei, Shandong, and Jiangsu, while the other is situated in the western part,
including Xinjiang, Tibet, and Yunnan provinces. Yunnan, rich in forest resources, bears the
brunt of El Niño 's impact, exhibiting a total negative GPP anomaly of $-90.21$ Tg C yr$^{-1}$ (Table
S4) and a relative change of approximately 16%. Despite comparable relative changes in GPP
for other provinces, their GPP anomalies are relatively smaller, ranging from $-10$ to $-15$ Tg C
yr$^{-1}$. Notably, Xinjiang, characterized by a fragile forest steppe in the Altai and Tianshan
Mountain regions, consistently demonstrates substantial relative changes in GPP during both
ENSO and other events. Quantitatively, during the El Niño episode, Xinjiang witnesses a
remarkable 24% relative change in GPP, accompanied by a positive GPP anomaly of 15.27 Tg
C yr$^{-1}$. In contrast, during the La Niña episode, provinces with notable relative changes are
mainly concentrated in the northern regions, such as Xinjiang, Inner Mongolia, Ningxia,
Shanxi, and Liaoning provinces (Fig. 7b). In addition, although the influence of ENSO on GPP
in the southern China is significant (Fig. 4), the total relative change through the year remains
small due to the cancellation of positive and negative anomalies in different seasons.

In the pIOD classification, only the 2019 extreme event is considered, resulting in the relative





change in GPP anomalies exceeding 10% in approximately half of the provinces. Notably,
Jiangxi, Fujian, Guangxi, Guangdong, and Hainan experience reductions of more than 25% in
GPP, with Jiangxi exhibiting the largest GPP anomaly of $-130$ Tg C yr$^{-1}$, Conversely,
Shandong, Shanxi, and Henan witness increase of over 25% in GPP (Fig. 7c). During nIOD
events, northern provinces generally exhibit negative relative changes, while southern
provinces display positive relative changes.

In summary, the relative changes in total GPP anomalies at the provincial scale exhibit an east-
west pattern in annual variation. Meanwhile, the influence of IOD events on GPP presents an
opposing north-south pattern.



**4. Discussion**
**4.1 The effect of compound ENSO and IOD events on China's GPP**

Fig. 8. Spatial distributions of seasonal composite GPP anomalies for compound events, (a-d) for El Niño & pIOD events, and (e-h) for La Niña & nIOD events.
The two-digit year in first column denote the years used for composite analysis.

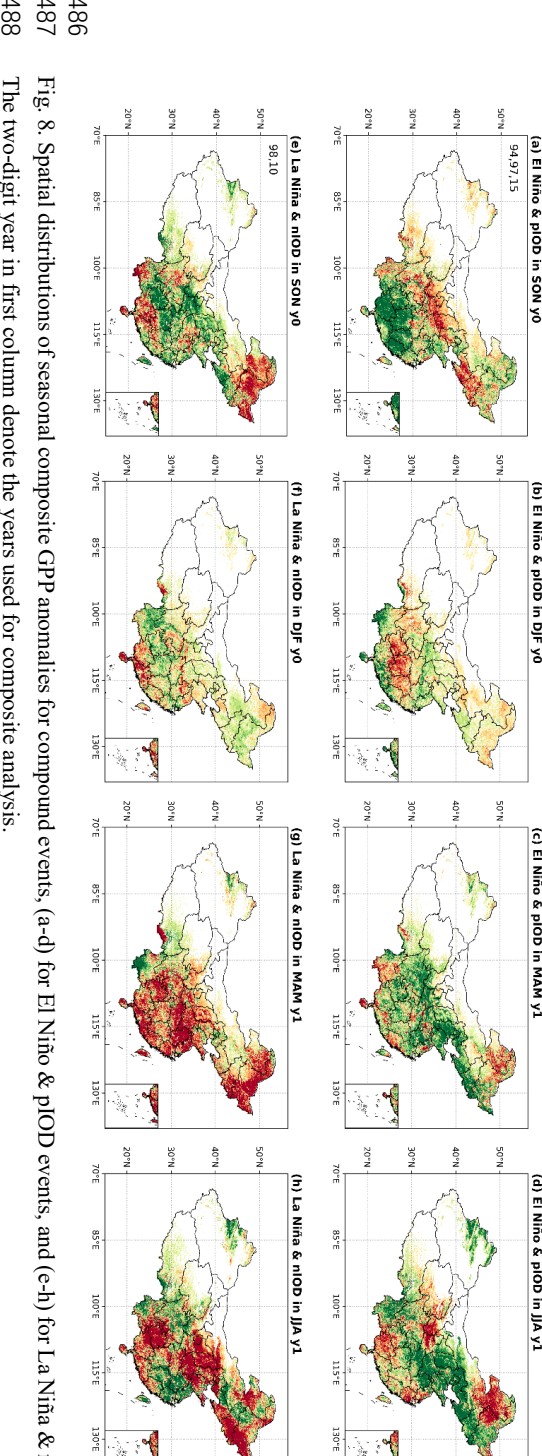



Indeed, despite IOD events being generally considered an independent coupled ocean-
atmosphere interaction (Saji et al., 1999), historical IOD events can occur in conjunction with
ENSO (Ham et al., 2017; Yang et al., 2015). These combined phenomena are most notable
represented by El Niño & pIOD and La Niña & nIOD events. Williams and Hanan (2011)
researched the interactive effects of ENSO and IOD on African GPP, relying on an offline
terrestrial biosphere model simulation. Their findings suggested that IOD could cause obvious
anomalous GPP over much of Africa, capable of suppressing or even reversing ENSO signals
in GPP anomalies. In addition, Yan et al. (2023) explored the interactive effects of ENSO and
IOD on seasonal anomalies of tropical net land carbon flux using the TRENDYv9 multi-model
simulations, revealing diverse effects in different sub-continents and seasons. We explore the
anomalies of GPP in compound events based on composite analysis (Fig. 8), and the spatial
patterns of soil moisture and temperature anomalies are shown in the appendix (Figs. S9 and
S10).

The spatial patterns of the GPP anomalies during concurrent ENSO and IOD events differ from
those in single events, although some similarities are evident. GPP anomalies in El Niño &
pIOD and La Niña & nIOD events are generally opposite, and we focus specifically on El Niño
& pIOD events here. In El Niño & pIOD events, GPP anomalies exhibit a general opposition,
with enhanced vegetation photosynthesis in the southern regions and inhibited in the northern
regions during SON. This spatial characteristic of GPP anomalies bears some resemblance to
that induced by El Niño alone (Figs. 4a and 8a). Weak GPP anomalies are generally observed
in DJF, with noticeable negative GPP anomalies in Guizhou and Hunan, and some positive
GPP anomalies in regions south of 25°N (Fig. 8b). Notably during DJF, while significant
positive GPP anomalies occur in El Niño events (Fig. 4b), simultaneous pIOD events induce
significant negative GPP anomalies (Fig. 5b). When both events coincide, their impacts seem
to largely counterbalance each other, resulting in a more neutral GPP anomaly. In MAM, GPP
increases in Northern China (Fig. 8c). Subsequently, in JJA, vegetation photosynthesis
experiences a significant increase in the Northern and Yunnan provinces (Fig. 8d).



It is worth noting that the impacts of compound events on China's GPP may not follow a
straightforward linear superposition of the effects of two individual events. While their effects
are nearly opposite when occurring separately, the positive and negative effects on GPP may
be not simply cancelled each other out when they coincide. This complexity arises from the
simultaneous occurrence of two tropical air-sea interaction modes, leading to intricate effects
on mid-latitude circulations. Given the limited number of compound events, further exploration
is necessary to unravel the effects of ENSO and IOD on GPP in China.

**4.2 Modulation of large-scale circulations on China's GPP**

China's GPP is intricately influenced by atmospheric circulations and sea surface temperature
(Li et al., 2021; Ying et al., 2022). Ying et al. (2022) showed significant correlations between
seasonal GPP variation in China and climate phenomena such as ENSO, Pacific Decadal
Oscillation (PDO), and Arctic Oscillation (AO), based on the Residual Principal Component
analysis. Their research indicated that these identified SST and circulation factors could
account for 13%, 23% and 19% of the seasonal GPP variations in spring, summer and autumn,
respectively. And Li et al. (2021) proved that GPP response to El Niño varied with PDO phases
during the growing seasons of typical El Niño years. Although both studies emphasized the
impact of ENSO on China's GPP and explored the roles of PDO and AO, the IOD was notably
absent from their analyses. Contrastingly, our study sheds light on the significant influence of
the extreme positive phase of IOD in 2019, showing a substantial negative GPP anomaly in
southeastern China during SON, aligning with findings by Wang et al. (2021b). Moreover, the
integration of partial correlation and composite analysis in our study elucidates the
considerable impact of IOD on China's GPP within this context. Importantly, our research
underscores the temporal and spatial variability in the effects of IOD and ENSO on GPP across
different seasons and regions. This complexity in ocean–atmosphere teleconnections implies
that other climate oscillations, such as Polar/Eurasia (polarEA) and Atlantic Multidecadal
Oscillation (AMO), might also contribute to influencing China's GPP (Zhu et al., 2017), which





is still worthy of further analysis and research.

**4.3 Uncertainties in BEPS Simulations**
The simulation of China's GPP by BEPS is subject to several sources of uncertainty inherent
in the model's structure, parameterizations, processes, and input data (Chen et al., 2012; Chen
et al., 2017; He et al., 2021a; Liu et al., 2018; Wang et al., 2021a). Leaf Area Index (LAI), a
crucial input for the BEPS model, is derived from global remote sensing data that inherently
possess uncertainties in spatial distribution and trend changes. Previous studies have
highlighted significant uncertainties in simulating carbon budget of global terrestrial
ecosystems when employing different LAI remote sensing data (Chen et al., 2019; Liu et al.,
2018). Foliage clumping index which is used to separate sunlit and shaded LAI can also cause
some uncertainties in simulating GPP, because the current version of BEPS used the time-
invariant satellite-derived clumping index (Chen et al., 2012). Biases in meteorological drivers,
such as precipitation, can further result in considerable uncertainties in simulating terrestrial
carbon cycle. The choice of precipitation products, for instance, has been shown to yield
considerable differences in simulated net land-atmosphere carbon flux (Wang et al., 2021c).
Moreover, BEPS model, like other terrestrial biosphere models, lacks consideration for
vegetation adaptability to rising $CO_2$ concentration, potentially leading to an overestimation of
the fertilization effect on GPP. In addition, the accuracy of simulations over agricultural areas
is compromised in BEPS, as it only considers crops with a C3 photosynthetic pathway and
overlooks C4 crops (He et al., 2017; He et al., 2021b; Ju et al., 2006). Although BEPS simulated
GPP demonstrates relatively high consistency with the measured GPP of Yingke Station (CRO),
located in the northwest of China, its accuracy lacks validation over the extensive farmlands in
north and northeastern China where various crops are grown (Fig. S11). Agricultural operations,
particularly irrigation, which can significantly impact GPP, are not considered in BEPS. He et
al. (2021a) revealed extensive wetting signals over croplands in arid and semi-arid areas which
exerted strong impacts on GPP and evapotranspiration simulations in BEPS after assimilating



the Soil Moisture Active Passive (SMAP) soil moisture product. Furthermore, photosynthetic
key parameters, such as carboxylation capacity at 25°C ($V_{cmax,25}$), can largely determine the
performance in simulating GPP. After assimilating the solar-induced chlorophyll fluorescence
(SIF) from the Orbiting Carbon Observing Satellite-2 (OCO-2) to optimize $V_{cmax,25}$ of different
plant functional types (PFTs) in BEPS, previous studies suggested the improvements in
simulating GPP at regional and global scales to some extent (He et al., 2019; Wang et al.,
2021a).

## 579    5. Conclusion

In this paper, we used partial correlation coefficients and composite analysis to investigate the
impacts of ENSO and IOD events on China's GPP during 1981–2021. The partial correlation
results reveal that the effects of ENSO and IOD on GPP and related climate in China exhibit
distinct seasonal variations and are basically opposite. Specifically, during SON, significant
negative *pcor* between GPP and ENSO is observed over the Tibetan Plateau, southwestern
China, Loess Plateau, and Liaoning. In DJF, strongly positive *pcor* occurs over southern China,
weakening in the subsequent MAM, albeit with some enhancements in northern Hebei and
neighboring Inner Mongolia. The *pcor* then turns generally negative in JJA. In contrast,
significant positive *pcor* between GPP and IOD is noted in southwestern and Northeast China
during SON. Subsequently, widespread negative *pcor* appears during DJF, persisting
significantly in most western and northern regions during MAM. In JJA, the *pcor* becomes
significantly positive in southwestern, north and northeast China. Moreover, the correlation
coefficients between GPP and climate show that GPP anomalies are primarily dominated by
SM in JJA and SON, while temperature generally plays a more important role in in DJF and
MAM.

The composite analysis results validate the patterns of GPP anomalies observed in the partial
correlation. Generally, China's annual total GPP demonstrates modest positive anomalies in La



Niña and nIOD years, contrasting with minor negative anomalies in El Niño and pIOD years.
This results from the counterbalancing effects, with significantly greater GPP anomalous
magnitudes in DJF and JJA. Regionally, GPP anomalies fluctuate more in the Southern and
Northern regions. The GPP anomaly in the Southern region dominates the national GPP
variation, with the contribution of 68% to ENSO events and 46% to IOD events, respectively.
On the provincial scale, western and northern provinces in experience larger relative annual
variations during ENSO events, with magnitudes exceeding 10%, exhibiting a general east-
west pattern. Conversely, provinces in the southern and Northern China witness larger relative
changes during IOD events, showing an opposing north-south pattern. For instance, the 2019
extreme pIOD led to relative changes of over 25% in certain provinces in the south and north.



**Acknowledgement**
The calculations in this paper have been done on the computing facilities in the High Performance
Computing Center (HPCC) of Nanjing University. This study was supported by the Natural Science
Foundation of China (Grants 42141005), the Natural Science Foundation of Jiangsu Province, China
(BK20221449), and the National Key Scientific and Technological Infrastructure project "Earth System
Numerical Simulation Facility" (grant 2023-EL-ZD-00022).

**Conflict of Interest**
The authors declare no competing interests.

**Data Availability**
REA5 meteorological data are available at https://cds.climate.copernicus.eu/cdsapp#!/dataset/rean
alysis-era5-single-levels?tab=overview. The remote-sensing GLOBMAP LAI data is available at
https://zenodo.org/record/4700264#.YzvSYnZBxD8/. The carbon dioxide emissions data is availa
ble at https://gml.noaa.gov/webdata/ccgg/trends/co2/co2_mm_mlo.txt. Vegetation type data for B
EPS simulations is obtained from https://lpdaac.usgs.gov/products/mcd12q1v006/. Soil texture d
ata is available at https://data.tpdc.ac.cn/zh-hans/data/611f7d50-b419-4d14-b4dd-4a944b141175. S
oil moisture and surface air temperature from ERA5-Land are available at https://cds.climate.c
opernicus.eu/cdsapp#!/dataset/reanalysis-era5-land-monthly-means?tab=overview. Sea surface temp
erature dataset from ERSSTv5 is available at https://psl.noaa.gov/data/gridded/data.noaa.ersst.v5.
html. Eight sites of the ten are from ChinaFlux (http://www.chinaflux.org/enn/index.aspx), and
two are from National Tibetan Plateau Third Pole Environment (http://data.tpdc.ac.cn/zh-hans).
FluxSat GPP Version 2.2 are available at https://avdc.gsfc.nasa.gov/pub/tmp/FluxSat_GPP.



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
