# Peer review of "Distinct Impacts of El Niño-Southern Oscillation and Indian Ocean Dipole"

_EGUsphere, 2024_

## Author Comment (AC1)

Thank you for your kind help and arrangement for reviewing our manuscript. Point-by-point replies to the comments are addressed as below.

**Reviewer #1:**

Yan et al. provide comprehensive investigations on Gross Primary Production (GPP) under El Niño-Southern Oscillation (ENSO) and Indian Ocean Dipole (IOD) conditions in China. Seasonal differences in GPP responses to ENSO and IOD are especially well summarized. I believe this research can be used not only to support the carbon sink component in the carbon neutrality goal but also to support agricultural productivity in China. I only have several technical corrections before accepting this manuscript.

1. L32, L33: nIOD, pIOD introduced in Abstract. I can understand the meaning for (p for positive and n for negative), but I think it would be hard to understand from readers.

Reply: Thanks for your suggestion, I have corrected this expression by correcting pIOD to "positive IOD" and nIOD to "negative IOD". (L32-33)

2. Similarly, pIOD already used in L85 but it is lately introduced at L88. (positive IOD)

Reply: Thanks for your suggestion, I have introduced the pIOD at L85 to "positive IOD".

3. L112: ERA5 needs it's full name.

Reply: Thanks for your suggestion, I have corrected this expression. (L112)

4. L172, L173: ~ indicates approximately in English. It is not used to express range in English. Please replace it with –.

Reply: Thanks for your suggestion, I have corrected this expression. (L172, L173)

5. Table 1: I wondered why pIOD has just one year and nIOD has only three years. Does this mean IOD without ENSO? Then, you need to clarify.

Reply: In table1, we give the year of occurrence of 3 types of events: ENSO alone (El

Niño or La Niña), IOD alone (pIOD or nIOD), and both events simultaneously. And neither ENSO nor IOD events of positive and negative phases is completely symmetric, therefore, it is normal for pIOD event to occur only once and nIOD to occur three times.

6. L620: REA5 -> ERA5

Reply: Thanks for your suggestion, I have corrected this expression. (L620)

---

## Author Comment (AC2)

Thank you for your kind help and arrangement for reviewing our manuscript. Point-by-point replies to the comments are addressed as below.

**Reviewer #2:**

The manuscript is comprehensive and covers the key points of the study on the impacts of El Niño-Southern Oscillation (ENSO) and Indian Ocean Dipole (IOD) on Gross Primary Productivity (GPP) in China. It highlights the importance of analyzing the impact of ENSO and IOD on GPP in China and reveals that the impact varies by season. Additionally, it shows that the drivers of GPP are different across seasons. However, I have a few main concerns about this paper. My specific comments are as follows:

Line 105: Please consider adding a study area section in the methods part and justify why you chose China as the study area.

Reply: ENSO and IOD are atmospheric circulation events that occur in the tropics, but their effects are not confined to the tropics. Previous studies have shown that both ENSO and IOD have profound effects on climate or vegetation in China (L73-77, L83-91). However, the impact of these two events on China's vegetation and carbon cycle has not comprehensive till now. This paper intends to systematically explore the impact of these two events on our nation's GPP.

Line 113: Why were surface air temperature (TAS) and volumetric soil moisture (SM) selected? Please justify these choices.

Reply: ENSO and IOD affect climate, which in turn affects vegetation. Conventionally, the interannual variability of the terrestrial carbon cycle is more closely related with the temperatures and water (Ahlstrom et al., 2015; Wang et al., 2016; Jung et al.,2017). Therefore, we focus on the temperature and soil moisture to analyze the impact of ENSO and IOD on GPP in this study.

Line 118: Please add more details on how the layers of soil moisture are weighted and aggregated.

Reply: Four layers of soil water data are provided in ERA5-Land data, namely, volumetric soil water layer 1 (0-7 cm), volumetric soil water layer 2 (7-28 cm), volumetric soil water layer 3 (28-100 cm), and volumetric soil water layer 4 (100-289 cm). The data unit is $m^3$ $m^{-3}$. We obtained soil moisture data of 0-289cm by the following formula:

$$SM_{0-289cm} = (0.07*SM_{0-7cm} + 0.21*SM_{7-28cm}$$
$$+ 0.72*SM_{28-100cm} + 1.89*SM_{100-289cm})/2.89$$

Line 122: The spatial resolution of different remote sensing products varies. Have all the RS products been resampled? If so, what is the spatial resolution?

Reply: The spatial resolution of the ERA5-Land dataset was 0.1° × 0.1°, and the original resolution of the BEPS GPP was 0.0727° × 0.0727°. It was mentioned in Section 2.2 that we resampled all the data to 0.1° × 0.1° (L164).

Line 193: Please justify why you chose partial correlation analysis.

Reply: In section 3.1 and 3.2, we want to explore the relationship between ENSO or IOD events and GPP in China. The simplest and most effective method is to calculate the correlation coefficient between them. But considering that ENSO and IOD events may occur simultaneously, the result is not accurate if the Pearson's correlation coefficient is used. The partial correlation coefficient can remove the influence of IOD when calculating the relationship between ENSO and GPP. Therefore, we choose partial correlation coefficient to explore the relationship between ENSO or IOD events and GPP.

Line 233: The authors investigated the varying impacts of surface air temperature (TAS) and volumetric soil moisture (SM) on photosynthesis (GPP). The GPP was simulated using the Boreal Ecosystem Productivity Simulator model, which utilized meteorological data and evapotranspiration (ET) as inputs, both of which are closely linked with temperature and soil moisture, respectively. How did you avoid circular effects? Do these effects inflate the correlation coefficients?

Reply: In the BEPS model, ET is an output data rather than an input data, which is also explained in the manuscript (L139). At present, all models use meteorological data to simulate the carbon flux data, and the results actually include decadal, interannual and seasonal signals. In this study, we only analyzed the interannual signals of GPP for different seasons. In addition, although the input data of the model contains many meteorological variables, the predominant meteorological factors in different regions are not clear, and it is necessary to clarify the predominant factors in different regions through statistical analysis.

Line 240: In Figures 2.4a and 2.4e, please explain why the soil moisture does not align closely with the "pcor" in the east of Guangxi Province, east of Qinghai Province, and west of Sichuan Province. What other variables could affect this?

Reply: What is said here is that the spatial patterns of GPP and SM are consistent on the national scale. We are focusing on large scale descriptions here, but conclusions may differ on small regional scales. For example, the GPP in the regions you mentioned may have a greater correlation with temperature.

Line 260: The pattern correlation coefficient is -0.09 in MAM, which indicates that significant predictors are missing. Please discuss why the coefficient is low in MAM in the discussion part.

Reply: Thanks for your question. In reference to your suggestions, we have added some descriptions of this issue in the Discussion. (Section 4.1)

We think that phenology has an important influence on GPP, therefore, we removed the non-growing season areas where the seasonal average temperature is below 0, as in the study of Wang et al. (2021). In the ENSO event, the correlation coefficient between GPP and TAS increased from $-0.09$ to $-0.18$ in MAM. And the correlation coefficient between climate factors and GPP in MAM is indeed lower than that in other seasons. On one hand, spring is the beginning of the growing season, and there are a lot of croplands in China, so crop management may disturb the relationship between GPP and climate factors. On the other hand, the legacy effect of the climate in the previous season

has been widely demonstrated (Bastos et al., 2020; Bastos et al., 2021). ENSO events peaked in the DJF, and climate anomalies in DJF may have a large legacy effect on spring GPP, which was not considered in this study. Specifically, the temperature is significantly higher during DJF in El Niño events (positive phase ENSO event), which will promote the advance of the growing season, and then impact the vegetation in the following spring. In addition, Zhou et al. (Zhou et al., 2017) found that the correlation between the start time of the growing season and spring GPP was 0.82±0.1 in North America, confirming the importance of phenology to spring GPP. The changes of phenology and its effects on vegetation are complex issues that were not considered in this study, but will be explored in future studies.

**Reference:**

Ahlstrom, A., Raupach, M. R., Schurgers, G., Smith, B., Arneth, A., Jung, M., et al. (2015). The dominant role of semi-arid ecosystems in the trend and variability of the land CO2 sink. Science, 348(6237), 895-899.

Bastos, A., Ciais, P., Friedlingstein, P., Sitch, S. and Zaehle, S., 2020. Direct and seasonal legacy effects of the 2018 heat wave and drought on European ecosystem productivity. Science Advances, 6(24): eaba2724.

Bastos, A. et al., 2021. Vulnerability of European ecosystems to two compound dry and hot summers in 2018 and 2019. Earth System Dynamics, 12(4): 1015-1035.

Wang, S., Zhang, Y., Ju, W., Qiu, B. and Zhang, Z., 2021. Tracking the seasonal and inter-annual variations of global gross primary production during last four decades using satellite near-infrared reflectance data. Science of the total Enviroment, 755.

Zhou, S. et al., 2017. Dominant role of plant physiology in trend and variability of gross primary productivity in North America. Scientific Reports, 7.

---

## Author Response (AR2)

Thank you for your kind help and arrangement for reviewing our manuscript. All comments are constructive which have been taken into account fully when we prepared our revised manuscript. Point-by-point replies to the comments are addressed as below.

Line 233: The authors investigated the varying impacts of surface air temperature (TAS) and volumetric soil moisture (SM) on photosynthesis (GPP). The GPP was simulated using the Boreal Ecosystem Productivity Simulator model, which utilized meteorological data and evapotranspiration (ET) as inputs. Both of these inputs are closely linked with temperature and soil moisture, respectively. How did you avoid circular effects? Do these effects inflate the correlation coefficients?

I do not think the authors' response answered my question. They still did not explain the possible circular effects in the models.

**Reply:** Thank you for your comments. This question has been raised twice, so we wonder if there was a misunderstanding. We appreciate the opportunity to clarify further, and here is our additional explanation:

(1) As mentioned in the main text (Section: 2.1 Dataset Used), the BEPS model in this study was driven by satellite-derived LAI and meteorological data (such as temperature, precipitation, and downward solar radiation) to simulate GPP and ET. Therefore, ET is not an input in BEPS model and temperature is a direct driver.

(2) Correlation and regression are widely accepted statistical methods for analyzing the main drivers of GPP/NEP simulations (single model or TRENDY multi-model simulations) on an interannual scale (Zeng et al., 2005; Piao et al., 2013; Ahlstrom et al., 2015; Wang et al., 2016; Jung et al., 2017; Humphrey et al., 2018). Hence, there are no methodological problems.

(3) Regardless of temperature, as it is a direct driving factor, we admit that changes in soil moisture are indeed potentially related to temperature, which seems to be the "circular effect" you mentioned here. However, we prefer to use the terms "direct" and "indirect effect", that is, temperature has a direct effect on vegetation, and it can also affect vegetation by affecting soil moisture (indirect effect). Similarly, Humphrey et al. (2021) used the results of several Earth System Models (ESMs) to discuss the direct

and indirect effects of soil moisture on the inter-annual fluctuations of terrestrial carbon sinks. In response to this problem, we have added a few sentences to the discussion (**4.4 Limitations and Future work**), which are as follows: "*Finally, it is worth noting that climate factors often interact closely with one another. For example, soil moisture can influence changes in surface air temperature, and vice versa. As a result, in addition to direct effects, climate drivers may also impact vegetation through indirect pathways. Humphrey et al. (2021) discussed the direct and indirect effects of soil moisture on variations in terrestrial interannual carbon sinks—specifically, through its influence on temperature and vapor pressure deficit (VPD)—using simulations from four Earth System Models. This area of interaction warrants further investigation in future research.*"

Line 240: In Figures 2.4a and 2.4e, please explain why the soil moisture does not align closely with the "pcor" in the east of Guangxi Province, east of Qinghai Province, and west of Sichuan Province. What other variables could affect this?

Regarding this question, I was expecting explanations in the discussion section to address why some regions deviate from the main general pattern, but the authors did not provide this explanation.

**Reply:** Many thanks for your comments.

(1) In terms of climate factors, temperature and water (precipitation or soil moisture) have long been recognized as the primary climate factors driving the inter-annual fluctuations of GPP/NEP (Zeng et al., 2005; Piao et al., 2013; Ahlstrom et al., 2015; Wang et al., 2016; Jung et al., 2017; Humphrey et al., 2018). Of course, it is undeniable that it is also regulated by other climate factors, such as VPD and radiation. Given that this study includes two climate oscillations (ENSO and IOD) across four seasons, incorporating additional climate factors would complicate the analysis. Therefore, this study focuses on the roles of temperature and soil moisture in driving the inter-annual fluctuations of GPP across different seasons for simplicity.

(2) In Figure 2, we calculate pattern correlations at the national scale to identify the primary driver of GPP variations. We acknowledge possible mismatches between the

pcor patterns of GPP and TAS/SM in specific regions. This mismatch may result from the weak relationship between GPP, TAS/SM, and ENSO in certain areas, as well as the influence of other factors on GPP variation, as previously mentioned.

(3) For this aspect, we have added some discussion in **4.4 Limitations and Future work** as "*Additionally, Temperature and water (precipitation or soil moisture) have long been regarded as the main climate factors driving inter-annual fluctuations of GPP or NEP (Zeng et al., 2005; Piao et al., 2013; Ahlstrom et al., 2015; Wang et al., 2016; Jung et al., 2017; Humphrey et al., 2018). However, other factors, such as VPD and radiation, also play important roles. This may explain the occasional mismatch between GPP patterns and TAS/SM in certain regions in Figs. 2 and 3. Overall, although the dominant driving factors vary seasonally, TAS and SM capture GPP variations more effectively on a national scale.*"

**Reference**

Ahlstrom, A., Raupach, M. R., Schurgers, G., Smith, B., Arneth, A., Jung, M., et al. (2015). The dominant role of semi-arid ecosystems in the trend and variability of the land $CO_2$ sink. Science, 348(6237), 895-899.

Humphrey, V., Zscheischler, J., Ciais, P., Gudmundsson, L., Sitch, S., & Seneviratne, S. I. (2018). Sensitivity of atmospheric CO2 growth rate to observed changes in terrestrial water storage. Nature, 560(7720), 628-631.

Humphrey, V., Berg, A., Ciais, P., Gentine, P., Jung, M., Reichstein, M., et al. (2021). Soil moisture–atmosphere feedback dominates land carbon uptake variability. Nature, 592(7852), 65-69.

Jung, M., Reichstein, M., Schwalm, C. R., Huntingford, C., Sitch, S., Ahlstrom, A., et al. (2017). Compensatory water effects link yearly global land $CO_2$ sink changes to temperature. Nature, 541(7638), 516-520.

Piao, S., Sitch, S., Ciais, P., Friedlingstein, P., Peylin, P., Wang, X., et al. (2013). Evaluation of terrestrial carbon cycle models for their response to climate variability and to $CO_2$ trends. Global Change Biology, 2117–2132.

Wang, J., Zeng, N., & Wang, M. (2016). Interannual variability of the atmospheric $CO_2$ growth rate: roles of precipitation and temperature. Biogeosciences, 13(8), 2339-2352.

Zeng, N., Mariotti, A., & Wetzel, P. (2005). Terrestrial mechanisms of interannual $CO_2$ variability. Global Biogeochemical Cycles, 19(1), GB1016.